# Faithful Knowledge Distillation

## Abstract

Knowledge distillation (KD) has received much attention due to its success in compressing networks to allow for their deployment in resource-constrained systems. While the problem of adversarial robustness has been studied before in the KD setting, previous works overlook what we term the relative calibration of the student network with respect to its teacher in terms of soft confidences. In particular, we focus on two crucial questions with regard to a teacher-student pair: (i) do the teacher and student disagree at points close to correctly classified dataset examples, and (ii) is the distilled student as confident as the teacher around dataset examples? These are critical questions when considering the deployment of a smaller student network trained from a robust teacher within a safety-critical setting. To address these questions, we introduce a *faithful imitation* framework to discuss the relative calibration of confidences, as well as provide empirical and certified methods to evaluate the relative calibration of a student w.r.t. its teacher. Further, to verifiably align the relative calibration incentives of the student to those of its teacher, we introduce *faithful distillation*. Our experiments on the MNIST, Fashion-MNIST and CIFAR-10 datasets demonstrate the need for such an analysis and the advantages of the increased verifiability of faithful distillation over alternative adversarial distillation methods.

## 1 Introduction

The state-of-the-art performance of deep neural networks in a variety of different application areas has recently been fuelled by significant increases in their capacity (Brown et al., 2020; Bommasani et al., 2021). However, the increase in the size of networks has led to deployment issues for resource-constrained systems (Gou et al., 2021; Mishra & Marr, 2017) such as self-driving cars and small medical devices (Wang et al., 2018). Simply deploying smaller versions of these networks, trained as usual, tends to hurt performance.

Knowledge distillation (KD) helps to deal with this problem by distilling knowledge from a large, high-performing, expensive-to-run neural network into a smaller network (Gou et al., 2021; Hinton et al., 2015; Tung & Mori, 2019). This has been shown to improve the performance of smaller networks compared to standard training (Hinton et al., 2015), bringing the performance of larger networks to smaller networks which tend to be more efficient to run. However, students trained using standard KD are vulnerable to *adversarial attacks*, that is, to small input perturbations which cause networks to change their correct classification (Guo et al., 2019; Goldblum et al., 2020).

To mitigate this issue, previous works have studied the robustness of teacher and student networks separately, in particular, that distilling knowledge from a robust teacher improves the robustness of the distilled student (Goldblum et al., 2020). However, an important question arises: if we have a robust teacher, *can we find dataset examples where the teacher and student agree, yet under small perturbations then disagree?* Moreover, consider the case where the confidence of the deployed network will be used to make decisions - *is the distilled student as confident as the teacher in and around dataset examples?* As fig. 1 shows, small perturbations of an image can cause significant disagreements in classification, which we refer to as *adversarial disagreement examples*, and major differences in confidences between a teacher and its student. This motivates the need for (i) a framework to evaluate the robustness and confidence of the student with respect

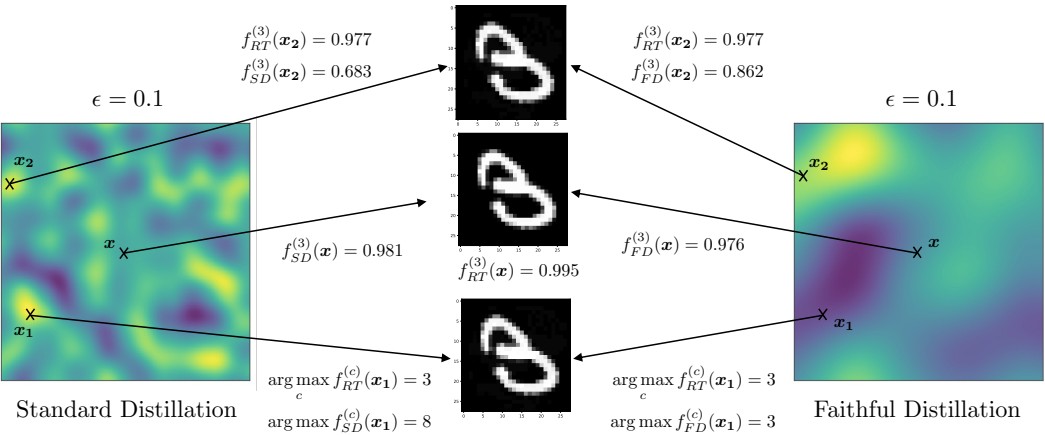

Figure 1: *Faithful Distillation* - a heat map showing the maximum difference (specifically the $\ell_\infty$-norm) of the soft confidence outputs of a robust teacher model ($f_{RT}$) and its two distilled students, a student trained using standard KD ($f_{SD}$), and a student trained using faithful distillation ($f_{FD}$) within an $\ell_\infty$-ball surrounding an example $\boldsymbol{x}$ from the MNIST dataset. In particular, we observe two problematic scenarios for $f_{SD}$: an adversarial disagreement example at $\boldsymbol{x_2}$, and poor relative calibration in terms of confidences (that is, large differences in confidence outputs of the two networks) at $\boldsymbol{x_1}$. We can see for $f_{FD}$, the heat map transitions are a lot smoother indicating smaller differences in confidences with its teacher. This implies that $f_{FD}$ more closely imitates its teacher than $f_{SD}$. Overall, we observe that faithful distillation improves upon the two problems observed with $f_{SD}$.

to the teacher and (ii) a training method that allows us to obtain better students with respect to that framework.

Our contributions are fourfold:

- we define the concept of a *faithful imitator* to discuss and bound the difference in confidences of a teacher and its student;

- we introduce empirical and verified methods to investigate and compute these bounds;

- we provide a faithful distillation loss that allows us to train students which are verifiably more aligned with their teacher network in terms of confidences, and

- we demonstrate the capabilities of our framework on the MNIST (LeCun et al., 1998), Fashion-MNIST (Xiao et al., 2017) and CIFAR-10 (Krizhevsky & Hinton, 2010) datasets.

## 2 Background and Related Work

Throughout this paper, we assume that we are dealing with a $C$-class classification problem and that $(\boldsymbol{x}, y) \sim \mathcal{D}$ is an example from a data distribution $\mathcal{D}$, with $\boldsymbol{x} \in \mathbb{R}^n$ an input vector and $y \in \{1, \ldots, C\}$ its associated class.

### 2.1 Knowledge Distillation

Knowledge distillation is the process of transferring information and representations from a larger teacher network, $f_t : \mathbb{R}^n \to [0, 1]^C$, to a usually significantly smaller student network, $f_s : \mathbb{R}^n \to [0, 1]^C$, with the standard aim of improving the performance of the student over regular training.

Hinton et al. (2015) popularised a specific distillation process that utilised the following loss:

$$\mathcal{L}_{SD}\left(\boldsymbol{x}, y, \Theta\right) = \alpha T^2 \mathcal{L}_{KL}\left(f_s(\boldsymbol{x}; T), f_t(\boldsymbol{x}; T)\right) + (1 - \alpha)\mathcal{L}_{CE}\left(f_s(\boldsymbol{x}; 1), y\right), \tag{1}$$

where $\mathcal{L}_{CE}$ refers to the standard cross-entropy loss function, and $f_t(\boldsymbol{x}; T)$ and $f_s(\boldsymbol{x}; T)$ refer to the softmax outputs with temperature scaling $T > 0$ for the teacher and student respectively evaluated at a point $\boldsymbol{x}$. We refer to distillation training using this loss as standard distillation (SD).

Goldblum et al. (2020) found that starting from a robust teacher network and only distilling knowledge using clean data can empirically result in the student network inheriting robustness from its teacher. However, the authors also noted that this is only sometimes the case and can depend on both the dataset and teacher.

## 2.2 Adversarial Attacks and Training

Adversarial examples are dataset examples that a network correctly classifies that can be deliberately perturbed to cause the network to change its classification (Goodfellow et al., 2015).

Several white-box (Goodfellow et al., 2015; Dong et al., 2018; Madry et al., 2018) and black box-techniques (Narodytska & Kasiviswanathan, 2017; Ilyas et al., 2018; Li et al., 2019) have been developed to generate adversarial examples. One such method utilises PGD attacks which make use of projective gradient ascent to maximise the training loss $\mathcal{L}$ within a constrained domain $\mathcal{S}$:

$$\boldsymbol{x}^{t+1} = \Pi_S\left(\boldsymbol{x}^t + \eta\,\text{sgn}\left(\boldsymbol{\nabla}_{\boldsymbol{x}}\mathcal{L}(\boldsymbol{x}^t, y, \Theta)\right)\right), \tag{2}$$

where $\eta$ denotes the step size of the gradient steps, $\Pi_S$ is the projection operator onto the set $S$, and sgn denotes the sign function (Madry et al., 2018). Usually, $\mathcal{S}$ is taken to be an $\ell_p$-ball surrounding a given example that we wish to attack, typically for $p = \infty$.

Several methods to defend against adversarial attacks have been introduced (Song et al., 2017; Samangouei et al., 2018; Guo et al., 2018). Such methods include adversarial training (Goodfellow et al., 2015; Madry et al., 2018; Na et al., 2018), which aims to train a network to be more robust to adversarial attacks. Madry et al. (2018) framed the problem of adversarial training the as the following saddle point problem:

$$\min_{\Theta} \mathbb{E}_{(\boldsymbol{x}, y) \sim \mathcal{D}}\left(\max_{\boldsymbol{\delta} \in \mathcal{S}} \mathcal{L}(\boldsymbol{x} + \boldsymbol{\delta}, y, \Theta)\right), \tag{3}$$

where the inner maximisation may be tackled using the PGD method described in eq. (2).

## 2.3 Adversarial Distillation Methods

Adversarially robust distillation (ARD) is a modification of the SD method discussed above (Goldblum et al., 2020). ARD is implemented so that the distilled student agrees with its teacher's unperturbed confidences even when the student itself is perturbed (Goldblum et al., 2020). The adversarial loss that is used within ARD is given by:

$$\mathcal{L}_{ARD}(\boldsymbol{x}, y, \Theta) = \alpha T^2 \mathcal{L}_{KL}\left(f_s(\boldsymbol{x} + \boldsymbol{\delta}_0; T), f_t(\boldsymbol{x}; T)\right) + (1 - \alpha)\mathcal{L}_{CE}\left(f_s(\boldsymbol{x}; 1), y\right), \tag{4}$$

where $\boldsymbol{\delta}_0 = \arg\max_{\|\boldsymbol{\delta}\| \leq \epsilon} \mathcal{L}_{CE}(f_s(\boldsymbol{x} + \boldsymbol{\delta}; 1), y)$ and KL denotes the KL divergence (Goldblum et al., 2020).

ARD training is used to distil both knowledge and robustness from a larger, more robust teacher network. The authors found that using ARD can experimentally produce more robust student networks that can even beat the performance in terms of robust accuracy of networks trained adversarially with the same network architecture (Goldblum et al., 2020).

Robust Soft Label Adversarial Distillation (RSLAD) is an adaption of ARD that replaces the hard labels within eq. (4) with soft labels given by the softmax outputs of the teacher (Zi et al., 2021). Specifically, the loss is given by

$$\mathcal{L}_{RSLAD}(\boldsymbol{x}, y, \Theta) = \alpha \mathcal{L}_{KL}\left(f_s(\boldsymbol{x} + \boldsymbol{\delta}_0; 1), f_t(\boldsymbol{x}; 1)\right) + (1 - \alpha)\mathcal{L}_{KL}\left(f_s(\boldsymbol{x}; 1), f_t(\boldsymbol{x}; 1)\right), \tag{5}$$

where $\boldsymbol{\delta}_0 = \arg\max_{\|\boldsymbol{\delta}\| \leq \epsilon} \mathcal{L}_{KL}\left(f_s(\boldsymbol{x} + \boldsymbol{\delta}; 1), f_t(\boldsymbol{x}; 1)\right)$.

Zi et al. (2021) found that RSLAD empirically produced more robust students than ARD with respect to both PGD attacks and even more sophisticated attacks such as AutoAttack (Croce & Hein, 2020).

## 3   Motivation

We motivate the need for a method of faithful distillation through the examples illustrated in fig. 1. Given a robust teacher network, we distil a student using the SD loss from section 2.1.

Let us consider an example of an image $\boldsymbol{x}$ from the MNIST dataset (LeCun et al., 1998). Both networks correctly classify the image $\boldsymbol{x}$ as a 3 with similar, high confidence levels. Let us now investigate perturbations of $\boldsymbol{x}$ in a neighbourhood of $\boldsymbol{x}$, specifically in an $\ell_\infty$-ball of radius $\epsilon = 0.1$.

We first find $\boldsymbol{x}_1$, a perturbation of the image $\boldsymbol{x}$, where the teacher classifies the perturbed image as a 3. However, the student classifies the perturbed image as an 8, despite the minimal difference between $\boldsymbol{x}_1$ and the original image $\boldsymbol{x}$. We will refer to an example where the student and teacher initially agree on a prediction but, under perturbation, the networks disagree with one another as an *adversarial disagreement example*. Formally, we define it as below:

**Definition 1 (Adversarial Disagreement Example)** *For an input $\boldsymbol{x} \in \mathbb{R}^n$ and teacher and student networks, $f_t : \mathbb{R}^n \to [0,1]^C$ and $f_s : \mathbb{R}^n \to [0,1]^C$ respectively, we say that $\boldsymbol{x}$ is an adversarial disagreement example for a given $\epsilon > 0$ if there exists an $\boldsymbol{x}' \in B_\epsilon(\boldsymbol{x})$ such that*

$$c_t(\boldsymbol{x}) = c_s(\boldsymbol{x}) \ and \ c_t(\boldsymbol{x}') \neq c_s(\boldsymbol{x}'), \tag{6}$$

*for $c_t(\boldsymbol{x}) = \arg\max_c f_t^c(\boldsymbol{x})$, $c_s(\boldsymbol{x}) = \arg\max_c f_s^c(\boldsymbol{x})$.*

We can view such examples as perturbations that cause an originally in agreement teacher-student pair to disagree. The quantity of such examples within a test set gives a measure of the robustness of the distillation process itself.

Next, we find $\boldsymbol{x}_2$, another perturbation of the image $\boldsymbol{x}$, where the teacher and student classify $\boldsymbol{x}_2$ as a 3, but with very different confidences in their predictions. Here, we would say that *the student is not relatively well calibrated in terms of confidence* to its teacher. This poses significant problems in the confident deployment of student networks in safety-critical environments where confidences are used in decision-making. In such a scenario, not only are correct classifications important, but also the confidence in said classification. Here, we see that even a small perturbation can result in wildly different confidences, which implies that such a deployed student poorly imitates and is not relatively well-calibrated to its teacher.

The above motivates the need for a framework in which to empirically investigate and verifiably confirm how relatively well-calibrated a student is to its teacher in terms of their confidences. Moreover, it calls for a form of faithful distillation that addresses the issues discussed above as measured within this new framework. Therefore, we present the faithful imitation framework in section 4 and a loss with the aim of producing verifiably more relatively calibrated students in section 5.

## 4   Faithful Imitator For Knowledge Distillation

In this section, we provide an evaluation framework based on the concept of faithful imitation, along with methods to compute lower and upper bounds based on it.

Assume we have two processes defined by the functions $f : \mathbb{R}^n \to \mathbb{R}^d$ and $\hat{f} : \mathbb{R}^n \to \mathbb{R}^d$, where the goal of $\hat{f}$ is to imitate the output of $f$. We define a *faithful imitator* as follows.

**Definition 2 (Faithful Imitator)** *We say that $\hat{f} : \mathbb{R}^n \to \mathbb{R}^d$ is an $(\epsilon, \delta)$- faithful imitation of $f : \mathbb{R}^n \to \mathbb{R}^d$ around $\boldsymbol{x}_0 \in \mathbb{R}^n$ if:*

$$d_f(f(\boldsymbol{x}), \hat{f}(\boldsymbol{x})) \leq \delta, \quad \forall \boldsymbol{x} \in B_\epsilon(\boldsymbol{x}_0) \tag{7}$$

where $B_\epsilon(\boldsymbol{x}_0) = \{\boldsymbol{x}' \in \mathbb{R}^n \mid d_{\boldsymbol{x}}(\boldsymbol{x}_0, \boldsymbol{x}') \leq \epsilon\}$, $d_f : \mathbb{R}^d \times \mathbb{R}^d \to \mathbb{R}_{\geq 0}$ *is a chosen metric function in the output space, and $d_{\boldsymbol{x}} : \mathbb{R}^n \times \mathbb{R}^n \to \mathbb{R}_{\geq 0}$ is a metric function in the input space. We refer to any $\delta$ that bounds $d_f(f(\boldsymbol{x}'), \hat{f}(\boldsymbol{x}'))$ as a faithfulness bound for a given $\epsilon$.*

Intuitively, this definition holds if for an $\epsilon$-neighbourhood defined by a function $d_{\boldsymbol{x}}$ around an input $\boldsymbol{x}_0$, the outputs of the imitator $\hat{f}$ and the original process $f$ are *similar* - up to a difference of $\delta$ - with respect to an output metric $d_f$.

Within the context of knowledge distillation, we have a student network, $f_s$, that is trying to imitate the output of a teacher network, $f_t$. We can use the definition of a faithful imitator as a principled way of reasoning about the *relative calibration* of a student network with respect to its teacher in terms of confidences. *By relative calibration, we refer to the similarity in the confidence outputs of the two networks.* For this purpose, we are interested, for a given $\epsilon$, in computing the tightest $\delta$ that satisfies definition 2. In the setting of multi-class classification, we follow the robustness literature and define $d_{\boldsymbol{x}}$ and $d_f$ to be the metrics induced from the $\ell_\infty$-norm (Madry et al., 2018).

For the sake of simplicity of our framework, within a $C$-class knowledge distillation setting, we assume $f_t : \mathbb{R}^{d_0} \to [0, 1]^C$ and $f_s : \mathbb{R}^{d_0} \to [0, 1]^C$ to be $L_t$ and $L_s$ fully connected neural networks, respectively, whose outputs are then normalised by a softmax function, $\sigma$, to yield the class confidences. That is, $f_t(\boldsymbol{x}) = \sigma(g_t(\boldsymbol{x}))$ and $f_s(\boldsymbol{x}) = \sigma(g_s(\boldsymbol{x}))$, where $g_t$ and $g_s$ are fully connected networks.

Formally, we define an $L$-layer fully connected neural network as a function $g : \mathbb{R}^{d_0} \to \mathbb{R}^{d_L}$, such that for an input $\boldsymbol{x} \in \mathbb{R}^{d_0}$, $g(\boldsymbol{x}) = \boldsymbol{z}^{(L)}(\boldsymbol{x}) = \boldsymbol{W}^{(L)}\boldsymbol{z}^{(L-1)}(\boldsymbol{x}) + \boldsymbol{b}^{(L)}$, where $\boldsymbol{z}^{(k)}(\boldsymbol{x}) = \boldsymbol{W}^{(k)}\phi(\boldsymbol{z}^{(k-1)}(\boldsymbol{x})) + \boldsymbol{b}^{(k)}$ for $k \in \{1, \ldots, L-1\}$, and $\boldsymbol{z}^{(0)}(\boldsymbol{x}) = \boldsymbol{x}$, in which $\boldsymbol{W}^{(k)} \in \mathbb{R}^{d_k \times d_{k-1}}$ and $\boldsymbol{b}^{(k)} \in \mathbb{R}^{d_k}$ are the weight and bias associated with the $k$-th layer of the network, and $\phi : \mathbb{R} \to \mathbb{R}$ is the element-wise activation function.

The computation of the optimal $\delta$ that satisfied definition 2 can be written in the form shown in eq. (8) below:

$$
\begin{aligned}
\max_{\boldsymbol{x} \in B_\epsilon(\boldsymbol{x}_0)} \quad & \|f_t(\boldsymbol{x}) - f_s(\boldsymbol{x})\|_\infty = \|\sigma(g_t(\boldsymbol{x})) - \sigma(g_s(\boldsymbol{x}))\|_\infty \\
\text{s.t.} \quad & g_t(\boldsymbol{x}) = \boldsymbol{W}_t^{(L_t)}\boldsymbol{z}_t^{(L_t-1)}(\boldsymbol{x}) + \boldsymbol{b}_t^{(L_t)} \\
& \boldsymbol{z}_t^{(k_t)}(\boldsymbol{x}) = \boldsymbol{W}_t^{(k_t)}\phi(\boldsymbol{z}_t^{(k_t-1)}(\boldsymbol{x})) + \boldsymbol{b}_t^{(k_t)} \\
& \boldsymbol{z}_t^{(0)}(\boldsymbol{x}) = \boldsymbol{x} \\
& \boldsymbol{g}_s(\boldsymbol{x}) = \boldsymbol{W}_s^{(L_s)}\boldsymbol{z}_s^{(L_s-1)}(\boldsymbol{x}) + \boldsymbol{b}_s^{(L_s)} \\
& \boldsymbol{z}_s^{(k_s)}(\boldsymbol{x}) = \boldsymbol{W}_s^{(k_s)}\phi(\boldsymbol{z}_s^{(k_s-1)}(\boldsymbol{x})) + \boldsymbol{b}_s^{(k_s)} \\
& \boldsymbol{z}_s^{(0)}(\boldsymbol{x}) = \boldsymbol{x} \\
& k_t \in \{1, \ldots, L_t - 1\} \\
& k_s \in \{1, \ldots, L_s - 1\}.
\end{aligned}
\tag{8}
$$

When solved to optimality for $\boldsymbol{x} \in \mathbb{B}_\epsilon(\boldsymbol{x}_0)$, the $\delta^*$ that maximises the differences between the confidence outputs of the two networks implies $f_s$ is a $(\epsilon, \delta^*)$-faithful imitator of $f_t$ around $\boldsymbol{x}_0$, meaning we can deploy $f_s$ with a guarantee that the confidence outputs of the student are similar to the teacher's outputs.

The issue with solving eq. (8) to optimality is that this is a general non-linear optimisation problem. In the following sections, we approach the computation of $\delta^*$ in two different ways: using an empirical, best-effort optimiser that gives us a lower bound on $\delta^*$ and by over-approximating and linearising the problem in a similar fashion to Zhang et al. (2018) obtaining an upper bound on $\delta^*$. The use of upper and lower bounds on the solution to eq. (8) provides us with a tractable way of evaluating the faithfulness, that is the degree of imitation in terms of confidences, of a student network to its teacher.

## 4.1 Empirical Lower Bounds on $\delta^*$

We can empirically obtain a lower bound on the solution to eq. (8) by using PDG attacks as defined in eq. (2) to maximise $\|f_t(\boldsymbol{x}) - f_s(\boldsymbol{x})\|_\infty$. By using PGD to maximise the difference in confidences between the

student and teacher networks within the $\ell_\infty$-ball around a given dataset point $\boldsymbol{x}_0$, we obtain a best-effort lower bound on $\delta^*$. This is useful for empirically investigating how large the difference in confidences of the teacher and student can be, but gives us no guarantee on the maximum possible difference within an $\epsilon$-ball surrounding the given image $\boldsymbol{x}$, and therefore cannot be considered a faithfulness bound.

### 4.2 Faithfulness Upper Bounds on $\delta^*$

To compute faithfulness bounds as per definition 2, we over-approximate the solution of eq. (8) by relaxing the problem using a linear formulation. We achieve this by (i) relaxing the non-linear activation functions $\phi$ at each layer using linear lower and upper bounds and (ii) relaxing the softmax $\sigma$ that yields $f_t(\boldsymbol{x}) = \sigma(g_t(\boldsymbol{x}))$ (and similarly for the student network).

Assuming that $\boldsymbol{z}_L^{(k)} \leq \boldsymbol{z}^{(k)} \leq \boldsymbol{z}_U^{(k)}$ for a given $k$ and an activation function $\phi$, we compute the parameters $\boldsymbol{\alpha}_L^{(k)}(\boldsymbol{z}^{(k)} + \boldsymbol{\beta}_L^{(k)}) \leq \phi(\boldsymbol{z}^{(k)}) \leq \boldsymbol{\alpha}_U^{(k)}(\boldsymbol{z}^{(k)} + \boldsymbol{\beta}_U^{(k)})$. For $\phi$ a ReLU activation, we can use the relaxations provided in Ehlers (2017) and Zhang et al. (2018) to obtain the parameters.

For an input vector $\boldsymbol{z} \in \mathbb{R}^C$, the softmax operator outputs a vector where the $i$-th component is defined as $\sigma(\boldsymbol{z})_i = \exp(z_i)/\sum_j \exp(z_j)$. Note that the softmax can be thus be written as $\sigma(\boldsymbol{z})_i = 1/(\sum_{j \neq i} \exp(z_j - z_i) + 1)$. Using this, we bound the $i$-th component of the softmax activation by first bounding the difference between logits. Specifically, given upper and lower bounds on the logits $z_i^{l_b} \leq z_i \leq z_i^{u_b}$ we compute the following over this domain:

$$d_{i,j}^{lb} = \min_{\boldsymbol{z}}(z_j - z_i) \;,\;\; d_{i,j}^{u_b} = \max_{\boldsymbol{z}}(z_j - z_i), \tag{9}$$

giving $d_{i,j}^{l_b} \leq z_j - z_i \leq d_{i,j}^{u_b}$ over inputs $\boldsymbol{z}$ whose components satisfy $z_i^{l_b} \leq z_i \leq z_i^{u_b}$. To obtain bounds on the $i$-th component of the softmax for each network, we propagate the logit difference bounds through the softmax function:

$$\sigma_i^{l_b} = \frac{1}{1 + \sum_{j \neq i} \exp\left(d_{i,j}^{u_b}\right)} \;,\; \sigma_i^{u_b} = \frac{1}{1 + \sum_{j \neq i} \exp\left(d_{i,j}^{l_b}\right)}. \tag{10}$$

This gives us $\sigma_i^{l_b} \leq \sigma(\boldsymbol{z})_i \leq \sigma_i^{l_b}$ for $z_i^{l_b} \leq z_i \leq z_i^{u_b}$.

We can apply the bounding of the softmax activation to $f_t$ and $f_s$ by using bounds on their logits, $z_{t,i}^{(L_t)}$ and $z_{s,i}^{(L_s)}$ respectively, to obtain upper and lower bounds of their soft outputs, $\boldsymbol{\sigma}^{u_b}(f_t(\boldsymbol{x}))$ and $\boldsymbol{\sigma}^{l_b}(f_s(\boldsymbol{x}))$. We can then compute a faithfulness upper on $\delta^*$ given by:

$$\max_{\boldsymbol{x} \in B_\epsilon(\boldsymbol{x}_0)} \|f_t(\boldsymbol{x}) - f_s(\boldsymbol{x})\|_\infty \leq \max\left(\left\|\boldsymbol{\sigma}^{u_b}(f_t(\boldsymbol{x})) - \boldsymbol{\sigma}^{l_b}(f_s(\boldsymbol{x}))\right\|_\infty, \left\|\boldsymbol{\sigma}^{u_b}(f_s(\boldsymbol{x})) - \boldsymbol{\sigma}^{l_b}(f_t(\boldsymbol{x}))\right\|_\infty\right). \tag{11}$$

For small enough $f_t$ and $f_s$, using a MILP solver directly, such as Gurobi (Gurobi Optimization, LLC, 2022), yields tighter bounds following the description above within reasonable runtimes compared to alternative bound propagation methods such as CROWN (Zhang et al., 2018).

**Extension to Convolutional Nerual Networks** One can extend the formulation introduced in eq. (8) and the bounding methods introduced above to networks such as convolutional neural networks. In particular, since a convolution is a linear operation, this can be bounded in a similar fashion to a standard fully-connected layer. We can further simply bound max-pooling layers as follows: given upper and lower bounds on the input neurons of a max pooling layer $\boldsymbol{z}_L^{(k)} \leq \boldsymbol{z}^{(k)} \leq \boldsymbol{z}_U^{(k)}$, we obtain upper and lower bounds on the max pooling layer by simply propagating these bounds through the max-pooling function MaxPool so that $\mathrm{MaxPool}(\boldsymbol{z}_L^{(k)}) \leq \mathrm{MaxPool}(\boldsymbol{z}^{(k)}) \leq \mathrm{MaxPool}(\boldsymbol{z}_U^{(k)})$.

**Connection to Model Calibration** Our proposed method of producing faithfulness bounds provides an upper bound on the relative calibration, that is the difference in confidences, of teacher and student networks. Therefore, a corollary of the introduced framework above is that if a teacher is robust and moreover is generally well-calibrated on a dataset, a distilled student that is a faithful imitator (i.e. achieves low empirical and faithfulness bounds) of its teacher will also be well-calibrated.

## 5 Faithful Distillation

We introduce a new form of distillation that we will refer to as *Faithful Distillation* (FD), which is defined by the following loss function:

$$\mathcal{L}_{FD}(\boldsymbol{x}, y, \Theta) = \alpha T^2 \mathcal{L}_{KL}\left(f_s(\boldsymbol{x} + \boldsymbol{\delta}_0; T), f_t(\boldsymbol{x} + \boldsymbol{\delta}_0; T)\right) + (1 - \alpha)\mathcal{L}_{CE}\left(f_s(\boldsymbol{x}; 1), y\right), \tag{12}$$

where $\boldsymbol{\delta}_0 = \arg\max_{\|\boldsymbol{\delta}\|_\infty \leq \epsilon} \mathcal{L}_{KL}\left(f_s(\boldsymbol{x} + \boldsymbol{\delta}), f_t(\boldsymbol{x} + \boldsymbol{\delta})\right)$.

This loss is an adaption of the ARD loss defined in eq. (4). Here, by minimising the loss, we aim to minimise the maximum difference in confidences between the student and its teacher over a given $\epsilon$ ball surrounding each input $\boldsymbol{x}$. This should encourage the student network to match its teacher's confidences even under perturbation, leading to a more relatively calibrated or better-imitating student.

It is worth noting that both FD and RLSAD should achieve comparable performance due to the fact that in the limit, they should both produce very similar bounds. This is because $f_t(x + \delta_0) \to f_t(x)$ as the teacher becomes increasingly robust during adversarial training. However, in section 6, we observe that on MNIST and Fashion-MNIST, FD seems to produce more verifiably relatively calibrated teacher-student pairs, which importantly provides a sound certificate for downstream tasks. Moreover, the increased verifiability is necessary in order to complement the less reliable empirical measures of faithfulness such as EMPLB introduced above. We discuss this topic in more depth in section 6.1.

## 6 Experiments

In this section, we apply the idea of faithful knowledge distillation to classification for MNIST (LeCun et al., 1998), Fashion-MNIST (F-MNIST) (Xiao et al., 2017) and CIFAR-10 (Krizhevsky & Hinton, 2010). We will investigate the robustness of individual models and of a given distillation process for $\ell_\infty$ neighbourhoods of radii $\epsilon \in \{0.025, 0.05, 0.1, 0.15, 0.2\}$ on MNIST and of radii $\epsilon \in \{4/255, 8/255, 12/255, 16/255, 20/255\}$ on F-MNIST and CIFAR-10. We explore how relatively well-calibrated a given student is to its teacher network by computing upper faithfulness bounds on $\delta^*$ as described in section 4.2. Furthermore, we will compute empirical lower bounds on $\delta^*$ as described in section 4.1 using PDG attacks of 50 iterations on each image with a step size of $2.5\epsilon/50$ (following Madry et al. (2018)), using 5 iterations of random restarts per image. For the sake of efficiency, we report results on a randomly sampled set of 1000 MNIST and F-MNIST images and 100 CIFAR-10 images from the original test sets - which we will refer to simply as the test set of each data set throughout.

**Experimental Setup.** As per section 4, both the teacher and student networks we train for MNSIT and F-MNIST are fully-connected, feed-forward networks that use ReLU activations. Specific network architecture details for networks trained on MNSIT and F-MNIST can be found in appendix A.1 and appendix A.2 respectively. The architectures chosen make the student networks around 37.7% and 52.5% smaller than their teacher network on MNIST and F-MNIST, respectively. On CIFAR-10, we utilise convolutional neural networks including convolutional, max-pooling, ReLU activations and fully-connected layers for both the teacher and students. Specific network architectures for the CIFAR-10 networks can be found in appendix A.3. Due to the network architectures used, on CIFAR-10, the student networks are around 48% smaller than their teacher. Note that all adversarial training for individual model training and for distillation to follow uses PGD augmentations (Madry et al., 2018) with $\epsilon = 0.2$ and a step size of 0.05 for 10 iterations for MNIST training, and $\epsilon = 8/255$ and a step size of $2/255$ for F-MNIST and CIFAR-10 training. Moreover, on CIFAR-10, we perform additional ablation studies to investigate the effect of the loss weighting used for FD training. These additional experiments can be founding in appendix A.4.

We start by training robust teacher networks on each dataset, which we denote by $f_{RT}$ as we would like to investigate and produce robust and relatively well-calibrated students distilled from these robust teachers. This is in line with the observation made by Goldblum et al. (2020) that robust teacher networks are better able to produce robust student networks. Note that all the students to follow are distilled from these particular teacher networks on their respective datasets. The teacher networks achieved test set accuracies of 97.88%, 88.23% and 63.09% on MNIST, F-MNIST and CIFAR-10 respectively.

From these teacher networks, we train four separate student networks on each dataset according to the distillation methods previously discussed. We train students using SD ($f_{SD}$), students using ARD ($f_{ARD}$), students using RSLAD ($f_{RSLAD}$) and students using our loss, FD ($f_{FD}$). We use an even weighting of loss terms in each distillation loss used for training. These networks achieved clean test set accuracies of 97.41%, 97.20%, 97.12% and 97.14% on MNIST, 88.29%, 88.18%, 87.47% and 87.81% on F-MNIST, and 63.09%, 62.28%, 60.55% and 60.48% on CIFAR-10 respectively. Further details of network training can be found in appendix A.1, appendix A.2 and appendix A.3.

**Measuring Robustness and Faithfulness.** To compare the performance of the four different students with respect to their robust teacher $f_{RT}$ on the two datasets, we are interested in measuring robustness and faithfulness.

For robustness, we analyse both the teacher and students alone by computing each model's robust accuracy using 50-step PGD attacks at different $\epsilon$ values with a step size of $2.5\epsilon/50$ (Madry et al., 2018; Goldblum et al., 2020). Further, to understand the additional errors the students make compared to the robust teacher - as per the motivation in section 3 and definition 1- we introduce the concept of distillation agreement, which is simply the percentage of examples from a dataset that do not cause disagreements between teacher and student under perturbation.

**Definition 3 (Distillation Agreement)** *For a dataset $\mathcal{D}$, teacher and student networks, $f_t$ and $f_s$ respectively, and $\epsilon > 0$, we define the set of original agreements, $\mathcal{S}_{OA} \subset \mathcal{D}$, and the set of adversarial disagreement examples, $\mathcal{S}_{ADE,\epsilon} \subset \mathcal{D}$, as*

$$\mathcal{S}_{OA}^{t,s} = \{\boldsymbol{x} \in \mathcal{D} : c_s(\boldsymbol{x}) = c_t(\boldsymbol{x})\} \quad and \quad \mathcal{S}_{ADE,\epsilon}^{t,s} = \{\boldsymbol{x} \in \mathcal{S}_{OA}^{t,s} : \exists \boldsymbol{x}' \in B_\epsilon(\boldsymbol{x}) \ s.t. \ c_t(\boldsymbol{x}') \neq c_s(\boldsymbol{x}')\}, \quad (13)$$

*for $c_t(\boldsymbol{x}) = \arg\max_c f_t^c(\boldsymbol{x})$, $c_s(\boldsymbol{x}) = \arg\max_c f_s^c(\boldsymbol{x})$. We then define the distillation agreement, $\mathcal{A}_\epsilon^t(f_s)$, of the student $f_s$ w.r.t. its teacher $f_t$ on $\mathcal{D}$ for a given $\epsilon > 0$ as:*

$$\mathcal{A}_\epsilon^t(f_s) = \frac{1}{|\mathcal{D}|} \sum_{\boldsymbol{x} \in \mathcal{D}} \left( \mathbb{1}_{\mathcal{S}_{OA}^{t,s}}(\boldsymbol{x}) - \mathbb{1}_{\mathcal{S}_{ADE,\epsilon}^{t,s}}(\boldsymbol{x}) \right), \quad (14)$$

*where as usual, $|\mathcal{D}|$ denotes the size of the set $\mathcal{D}$.*

Since $S_{ADE,\epsilon}^{t,s}$ is hard to compute exactly due to the nature of $\boldsymbol{x}' \in B_\epsilon(\boldsymbol{x})$, we approximate it using 50-step PGD attacks, again with a step size of $2.5\epsilon/50$, maximising the cross entropy loss of the student compared against the hard predictions of its teacher, and report instead *empirical distillation agreement, $\tilde{\mathcal{A}}_\epsilon^t(f_s)$.*

## 6.1 Results

**Robustness.** The empirical distillation agreement with respect to the teacher, $\tilde{\mathcal{A}}_\epsilon^{RT}$, and robust accuracies for each of the networks on the MNIST, F-MNIST and CIFAR-10 datasets are shown in table 1.

We observe on both datasets that incorporating any form of adversarial distillation (ARD, RSLAD, or FD) produces a more robust student, as seen through the robust accuracies, than SD. The robust accuracy of $f_{FD}$, $f_{ARD}$ and $f_{RSLAD}$ are similar. However, we note that RSLAD achieves the greatest robust accuracy scores across all values of $\epsilon$ on both MNIST and F-MNIST. This agrees with the general findings of Zi et al. (2021) that using soft labels over hard labels helps produce more robust student networks.

Moreover, we see that the empirical distillation agreement scores of all adversarially distilled students on both datasets are significantly greater than for $f_{SD}$, indicating that adversarial training helps to align the predictions of teacher and student networks under perturbation. On MNIST, we observe that $f_{FD}$ obtains the greatest empirical distillation agreement over all but the smallest $\epsilon$. This holds also on the CIFAR-10. However, on the F-MNIST dataset, we observe that $f_{RSLAD}$ obtained the greatest empirical distillation agreement, with $f_{FD}$ achieving, on average, a higher agreement score than $f_{ARD}$. One potential cause for the difference in results could be the sensitivity of the trained networks to hyperparameter choice. Future work could investigate this or other reasons for the differences in adversarial disagreement scores across datasets.

Table 1: *Robustness* - robust accuracies and empirical distillation agreements on MNIST, F-MNIST and CIFAR-10.

| | $\epsilon$ | Individual Model Robust Accuracy (%) | | | | | Empirical Distillation Agreement (%) | | | |
|---|---|---|---|---|---|---|---|---|---|---|
| | | $f_{RT}$ | $f_{SD}$ | $f_{ARD}$ | $f_{RSLAD}$ | $f_{FD}$ | $\tilde{\mathcal{A}}^{RT}_{\epsilon}(f_{SD})$ | $\tilde{\mathcal{A}}^{RT}_{\epsilon}(f_{ARD})$ | $\tilde{\mathcal{A}}^{RT}_{\epsilon}(f_{RSLAD})$ | $\tilde{\mathcal{A}}^{RT}_{\epsilon}(f_{FD})$ |
| MNIST | 0 | 97.9 | 97.3 | 96.6 | 97.0 | 97.0 | 98.4 | 97.7 | 98.0 | 98.3 |
| | 0.025 | 97.5 | 95.1 | 95.9 | **96.5** | 95.9 | 96.3 | 96.7 | **97.7** | 97.3 |
| | 0.05 | 96.7 | 94.1 | 94.7 | **95.7** | 95.5 | 95.5 | 95.7 | 96.9 | **97.0** |
| | 0.1 | 94.9 | 88.7 | 93.0 | **93.8** | 93.4 | 90.2 | 94.3 | 95.1 | **95.6** |
| | 0.15 | 92.9 | 80.3 | 89.9 | **90.8** | **90.8** | 82.2 | 92.2 | 92.7 | **93.9** |
| | 0.2 | 90.0 | 65.6 | 86.7 | **87.0** | 86.0 | 68.7 | 90.0 | 90.3 | **91.1** |
| F-MNIST | 0 | 89.1 | 89.1 | 89.1 | 88.4 | 88.6 | 95.3 | 95.9 | 97.3 | 95.9 |
| | 4/255 | 86.5 | 81.2 | 85.5 | **85.9** | 85.5 | 86.8 | 92.8 | **94.9** | 93.1 |
| | 8/255 | 83.8 | 72.3 | 82.9 | **83.5** | 82.4 | 76.4 | 91.7 | **93.1** | 90.3 |
| | 12/255 | 80.9 | 61.2 | 78.2 | **79.0** | 77.0 | 66.2 | 87.3 | **89.3** | 85.0 |
| | 16/255 | 76.9 | 49.1 | 72.2 | **74.5** | 72.2 | 54.6 | 80.4 | **85.7** | 81.9 |
| | 20/255 | 72.2 | 37.7 | 67.3 | **68.6** | 67.1 | 43.6 | 77.6 | **81.7** | 78.8 |
| CIFAR-10 | 0 | 72.0 | 70.0 | 68.0 | 65.0 | 74.0 | 79.0 | 79.0 | 81.0 | 89.0 |
| | 4/255 | 68.0 | 53.0 | 60.0 | 59.0 | **69.0** | 60.0 | 67.0 | 76.0 | **80.0** |
| | 8/255 | 58.0 | 36.0 | 53.0 | 48.0 | **56.0** | 42.0 | 61.0 | 62.0 | **71.0** |
| | 12/255 | 46.0 | 27.0 | 40.0 | 38.0 | **42.0** | 31.0 | 47.0 | 52.0 | **57.0** |
| | 16/255 | 35.0 | 14.0 | 29.0 | 30.0 | **37.0** | 20.0 | 38.0 | 44.0 | **55.0** |
| | 20/255 | 27.0 | 5.0 | 21.0 | 21.0 | **25.0** | 13.0 | 31.0 | 34.0 | **43.0** |

Table 2: *Faithfulness and relative calibration* - empirical lower bounds (EMPLB) and faithfulness upper bounds (FAITHUB) on MNIST, F-MNIST and CIFAR-10. *Lower is better.*

| | $\epsilon$ | EMPLB | | | | FAITHUB | | | |
|---|---|---|---|---|---|---|---|---|---|
| | | $f_{SD}$ | $f_{ARD}$ | $f_{RSLAD}$ | $f_{FD}$ | $f_{SD}$ | $f_{ARD}$ | $f_{RSLAD}$ | $f_{FD}$ |
| MNIST | 0.025 | $0.042_{\pm0.102}$ | $0.045_{\pm0.113}$ | $0.039_{\pm0.087}$ | $\mathbf{0.033}_{\pm0.079}$ | $0.073_{\pm0.150}$ | $0.061_{\pm0.143}$ | $0.060_{\pm0.118}$ | $\mathbf{0.054}_{\pm0.113}$ |
| | 0.05 | $0.060_{\pm0.129}$ | $0.055_{\pm0.130}$ | $0.049_{\pm0.101}$ | $\mathbf{0.041}_{\pm0.091}$ | $0.179_{\pm0.257}$ | $0.097_{\pm0.198}$ | $0.101_{\pm0.175}$ | $\mathbf{0.094}_{\pm0.171}$ |
| | 0.1 | $0.106_{\pm0.186}$ | $0.078_{\pm0.164}$ | $0.072_{\pm0.129}$ | $\mathbf{0.061}_{\pm0.117}$ | $0.731_{\pm0.343}$ | $0.262_{\pm0.337}$ | $0.290_{\pm0.319}$ | $\mathbf{0.248}_{\pm0.305}$ |
| | 0.15 | $0.172_{\pm0.246}$ | $0.106_{\pm0.197}$ | $0.101_{\pm0.160}$ | $\mathbf{0.086}_{\pm0.145}$ | $0.974_{\pm0.123}$ | $0.636_{\pm0.383}$ | $0.668_{\pm0.355}$ | $\mathbf{0.588}_{\pm0.367}$ |
| | 0.2 | $0.258_{\pm0.302}$ | $0.140_{\pm0.229}$ | $0.138_{\pm0.192}$ | $\mathbf{0.118}_{\pm0.174}$ | $0.999_{\pm0.017}$ | $0.905_{\pm0.230}$ | $0.889_{\pm0.229}$ | $\mathbf{0.884}_{\pm0.238}$ |
| F-MNIST | 4/255 | $0.099_{\pm0.159}$ | $0.067_{\pm0.112}$ | $\mathbf{0.053}_{\pm0.094}$ | $0.060_{\pm0.099}$ | $0.215_{\pm0.267}$ | $0.128_{\pm0.182}$ | $\mathbf{0.118}_{\pm0.170}$ | $0.123_{\pm0.171}$ |
| | 8/255 | $0.155_{\pm0.212}$ | $0.096_{\pm0.145}$ | $\mathbf{0.081}_{\pm0.130}$ | $0.089_{\pm0.131}$ | $0.617_{\pm0.393}$ | $0.330_{\pm0.342}$ | $0.341_{\pm0.341}$ | $\mathbf{0.322}_{\pm0.334}$ |
| | 12/255 | $0.219_{\pm0.265}$ | $0.129_{\pm0.179}$ | $\mathbf{0.112}_{\pm0.163}$ | $0.122_{\pm0.165}$ | $0.892_{\pm0.267}$ | $0.678_{\pm0.381}$ | $0.690_{\pm0.370}$ | $\mathbf{0.668}_{\pm0.380}$ |
| | 16/255 | $0.297_{\pm0.310}$ | $0.165_{\pm0.214}$ | $\mathbf{0.148}_{\pm0.197}$ | $0.160_{\pm0.199}$ | $0.990_{\pm0.071}$ | $0.906_{\pm0.235}$ | $0.917_{\pm0.214}$ | $\mathbf{0.902}_{\pm0.237}$ |
| | 20/255 | $0.378_{\pm0.343}$ | $0.204_{\pm0.243}$ | $\mathbf{0.190}_{\pm0.226}$ | $0.202_{\pm0.229}$ | $0.999_{\pm0.013}$ | $0.986_{\pm0.089}$ | $0.988_{\pm0.079}$ | $\mathbf{0.985}_{\pm0.090}$ |
| CIFAR-10 | 4/255 | $0.282_{\pm0.145}$ | $0.242_{\pm0.139}$ | $0.226_{\pm0.126}$ | $\mathbf{0.187}_{\pm0.096}$ | $0.671_{\pm0.191}$ | $0.546_{\pm0.203}$ | $0.497_{\pm0.176}$ | $\mathbf{0.474}_{\pm0.182}$ |
| | 8/255 | $0.388_{\pm0.166}$ | $0.317_{\pm0.155}$ | $0.291_{\pm0.139}$ | $\mathbf{0.245}_{\pm0.114}$ | $0.983_{\pm0.033}$ | $0.910_{\pm0.127}$ | $0.873_{\pm0.133}$ | $\mathbf{0.864}_{\pm0.153}$ |
| | 12/255 | $0.489_{\pm0.175}$ | $0.394_{\pm0.169}$ | $0.357_{\pm0.148}$ | $\mathbf{0.302}_{\pm0.129}$ | $1.000_{\pm0.002}$ | $1.000_{\pm0.016}$ | $0.990_{\pm0.035}$ | $\mathbf{0.989}_{\pm0.038}$ |
| | 16/255 | $0.583_{\pm0.177}$ | $0.462_{\pm0.178}$ | $0.415_{\pm0.157}$ | $\mathbf{0.359}_{\pm0.142}$ | $1.000_{\pm0.000}$ | $1.000_{\pm0.002}$ | $\mathbf{0.999}_{\pm0.008}$ | $\mathbf{0.999}_{\pm0.009}$ |
| | 20/255 | $0.666_{\pm0.165}$ | $0.530_{\pm0.176}$ | $0.474_{\pm0.164}$ | $\mathbf{0.416}_{\pm0.151}$ | $1.000_{\pm0.000}$ | $1.000_{\pm0.000}$ | $\mathbf{1.000}_{\pm0.001}$ | $\mathbf{1.000}_{\pm0.001}$ |

**Faithfulness and Relative Calibration.** To understand the faithfulness – and therefore relative calibration of confidences – of the different students w.r.t. the robust teacher networks, we report aggregated values of empirical lower bounds (EMPLB) and verified faithfulness bounds (FAITHUB) over the test set, and for each $\epsilon$, for both MNIST and F-MNIST in table 2. To fully capture the distribution of the bounds over the test set, we additionally present the results in fig. 2 for MNIST and in fig. 3 for F-MNIST. Here, *lower is better, implying a greater degree of relative calibration between a teacher-student pair.*

From table 2, we observe that the empirical attack bounds (EMPLB) on both datasets are on average smaller for $f_{FD}$, $f_{RSLAD}$ and $f_{ARD}$ than for $f_{SD}$. This indicates that the adversarially distilled students are, on

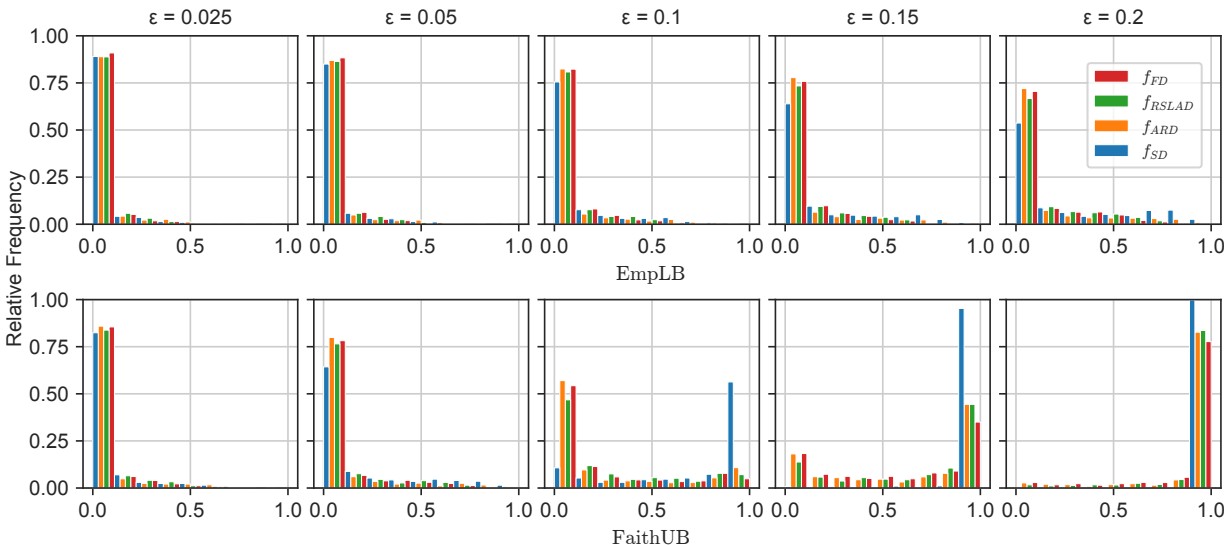

Figure 2: *Faithfulness and Relative Calibration* - histograms of the distributions of empirical lower bound (EmpLB) and faithfulness upper bounds (FaithUB) over the test set for different values of $\epsilon$, for students distilled from the robust teacher ($f_{RT}$) using SD($f_{SD}$), ARD ($f_{SD}$), RSLAD ($f_{RSLAD}$) and FD ($f_{FD}$) on MNIST.

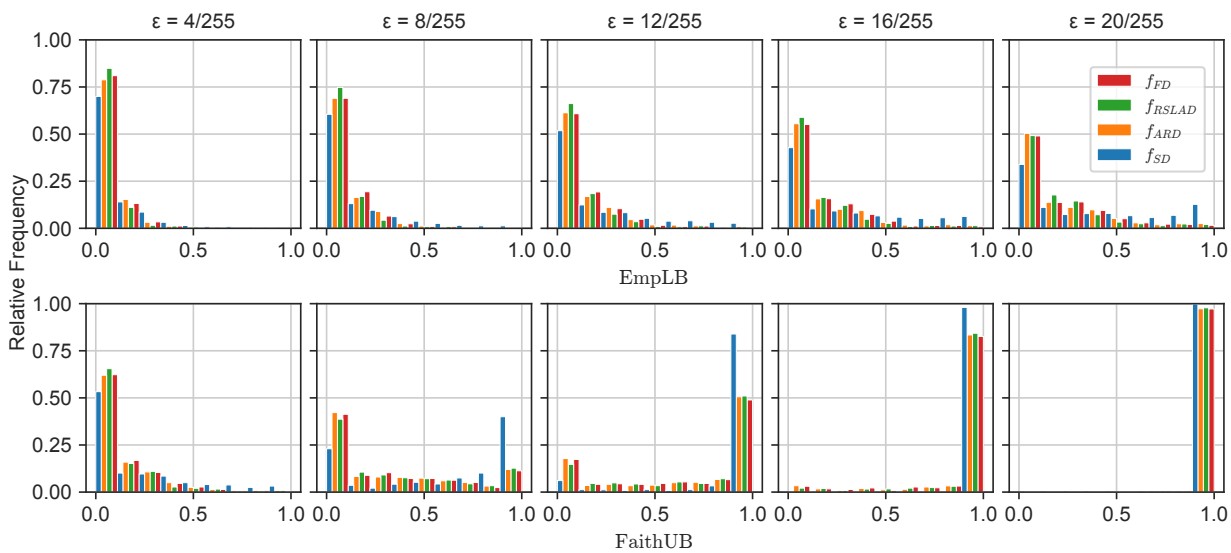

Figure 3: *Faithfulness and Relative Calibration* - histograms of the distributions of empirical lower bounds (EmpLB) and faithfulness upper bounds (FaithUB) over the test set for different values of $\epsilon$, for students distilled from the robust teacher ($f_{RT}$) using standard distillation ($f_{SD}$), ARD ($f_{SD}$), RSLAD ($f_{RSLAD}$) and FD ($f_{FD}$) on the F-MNIST data set.

average, *empirically more relatively well-calibrated than SD students* w.r.t. the robust teacher. This is exemplified further in fig. 2, where there is a greater shift in the empirical bound distributions for $f_{SD}$ over $f_{FD}$, $f_{RSLAD}$ and $f_{ARD}$. In addition, on the MNIST and CIFAR-10 datasets, we see that the empirical bounds are lower for $f_{FD}$ than for the three other student networks. Moreover, on MNIST and CIFAR-10, we notice the standard deviations of the bounds for $f_{FD}$ are smaller across all values of $\epsilon$, showing a more concentrated spread of lower empirical bounds than for the other student, with fewer images creating large

Table 3: Table showing EMPLB computed using PGD attacks with varying numbers of optimisation steps for three examples, $x_1$, $x_2$ and $x_3$, from the Fashion MNIST test set.

| | | EmpLB | | | |
| --- | --- | --- | --- | --- | --- |
| | | PGD-1 | PGD-5 | PGD-25 | PGD-50 |
| $x_1$ | RSLAD | **0.04679** | **0.06828** | 0.07177 | 0.07182 |
| | FD | 0.05060 | 0.06968 | **0.07091** | **0.07111** |
| $x_2$ | RSLAD | **0.1221** | 0.1401 | **0.1420** | **0.1426** |
| | FD | 0.1240 | **0.1381** | 0.1431 | 0.1439 |
| $x_3$ | RSLAD | **0.1289** | **0.1821** | **0.1861** | 0.1884 |
| | FD | 0.1879 | 0.1879 | 0.1879 | **0.1879** |

differences in the confidences of $f_{FD}$ and $f_{RT}$. This is further highlighted in the upper tails of the EMPLB distributions in fig. 2. On F-MNIST, like in the table 1, we again observe a change of results, with $f_{RSLAD}$ attaining the lowest empirical bounds followed by $f_{FD}$.

Looking at the faithfulness bounds (FAITHUB) for the $f_{FD}$, $f_{RSLAD}$ and $f_{ARD}$ students on both datasets in table 2, we observe that they are significantly lower across all values of $\epsilon$ than for $f_{SD}$, which confirms the empirical observations discussed above. Moreover, the higher standard deviation of the faithfulness bounds for $\epsilon$ values of 0.15 and 0.2 for $f_{FD}$, $f_{SD}$ and $f_{ARD}$ indicate that these students have less of an accumulation of images with faithfulness bounds of 1 - the largest possible difference in confidences between a teacher and its student. This is further shown on MNIST in fig. 2, where we observe empirically and verifiably that the maximum confidence difference between the adversarially distilled students and their teacher is smaller for a greater number of images, with their faithfulness bounds more closely imitating the empirical attack bounds for smaller values of $\epsilon$ than for $f_{SD}$. This suggests, in particular, that the degree of relative calibration between these adversarially trained students and their teacher is greater in a verifiable, upper-bound sense.

On all datasets, we see that as for nearly all values of $\epsilon$, we observe that the faithfulness bounds are, on average lower for $f_{FD}$ than for all of the other students across both datasets. *This shows that $f_{FD}$ is verifiably more relatively calibrated in an upper bound sense to its teacher on both MNIST, F-MNIST and CIFAR-10.* On MNIST and CIFAR-10, this aligns and supports the empirical observations that $f_{FD}$ is more faithful to its teacher in terms of confidences. On F-MNIST, we have a disparity where $f_{FD}$ is verifiably more relatively calibrated across nearly all of the values of $\epsilon$ as shown by lower verified bounds but empirically is observed to be less well relatively calibrated to its teacher than $f_{RSLAD}$, as shown by $f_{FD}$'s greater empirical faithfulness bounds and lower empirical distillation agreement.

Finally, we note that the standard deviations of the computed FAITHUB on all datasets tend to be large. This is a result of the fact that the bounds are bounded above by 1. In particular, this means that high variance in cases where the mean is higher is desirable. This is observable from the distribution of the MNIST bounds for $\epsilon = 0.15$ and 0.2 plotted in fig. 2, where we notice that the higher standard deviation of FD comes from having a greater accumulation of bounds in lower sections of the histograms.

**FD as a more verifiable distillation method over RLSAD.** As mentioned above, from table 1 and table 2, we see that the empirical measures of distillation robustness that we have introduced, that of EMPLB and empirical distillation agreement, are worse for $f_{FD}$ than for $f_{RSLAD}$ on F-MNIST. However, we note that the general trend of $f_{FD}$ producing tighter FAITHUB holds for all datasets.

These empirical measures, however, provide no guarantees on the relative calibration of a student to its teacher. Indeed, such methods are highly dependent on the method of attack used for their computation. In particular, table 3 shows three examples, $x_1$, $x_2$, and $x_3$, from the F-MNIST test set where, for $\epsilon = 8/255$, we compute EMPLBs using PGD attacks with steps of $1, 10, 25$, and $50$. We observe that $f_{RSLAD}$ and $f_{FD}$ flip between giving the lowest and hence the best EMPLB. This indicates that the aforementioned empirical methods cannot solely be used for comparisons. Moreover, combined with the difference in results across MNIST and F-MNIST, this highlights the importance of verifiable methods of bounding the maximum

difference in confidences between a teacher and student. In particular, FATIHUBs do provide certified upper bounds as they are computed using linear relaxations as a MILP and are therefore independent of and not subject to a choice of attack method such as PGD. As a result, FAITHUBs are a more robust method of evaluating the relative calibration of a teacher-student pair and, importantly, provide guarantees for downstream applications such as in safety-critical devices. It is still worth noting that a more complete picture is given when FAITHUB is evaluated alongside EMPLB.

### 6.2   Limitations

For larger values of $\epsilon$, the verified faithfulness bounds for all our student networks are loose compared to the empirically produced bounds. This indicates that our method of computing faithfulness bounds struggles to scale with increasing values of $\epsilon$. Since LP methods will provide tighter bounds than bound propagation methods such as CROWN (Zhang et al., 2018), this proves to be a limitation for verifying the greater degree of relative calibration seen empirically for larger values of $\epsilon$. This trend continues in particular to the larger CIFAR-10 dataset. This alludes to the fact that a more sophisticated method of producing faithfulness bounds needs to be produced to verify larger networks for larger values of $\epsilon$ on more complicated data sets.

## 7   Conclusion

The setting of faithful imitators provides a framework for empirically and verifiably reasoning about the relative calibration between a student network and its teacher in terms of confidences in a KD setting. This allows for guarantees on the maximum difference in confidences between the two networks, which is important for the safe deployment of student networks in safety-critical environments. Our experiments on MNIST, Fashion-MNIST and CIFAR-10 suggest that when combined with a robust teacher, ARD, RSLAD, and our FD training can produce empirically and verifiably relatively better-calibrated teacher-student network pairs than standard non-adversarial distillation training. Our FD-trained students were observed to be verifiably more relatively well-calibrated on average to their teacher network than standard non-adversarial distillation, RSLAD, or ARD students on both datasets. However, we observed that the results of empirical methods of bounding the difference in confidences varied across the two datasets for the adversarially trained students, highlighting the need for verifiable guarantees provided by our framework and methods.

Future work could explore the relative calibration between teacher-student pairs on larger datasets and further compare the relative calibration of teacher-student pairs trained using ARD, RSLAD, and FD across datasets. Moreover, more sophisticated and scalable ways of producing faithfulness bounds may be needed to develop this work further.

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

## A    Appendix

### A.1    Training Settings for Networks Used in MNIST Experiments

We train all models over 64 epochs using early stopping with a patience of 8 epochs based on the test accuracy. The teacher and student networks have {30, 30, 30, 30, 30} and {20, 20, 20, 20} neurons per fully-connected layer.

The teacher, $f_{RT}$, was trained using adversarial training, utilising the PGD attacks with an $\epsilon$ of 0.2 and step size of 0.05 for 10 iterations. We used SGD with a momentum of 0.9, a weight decay factor of 0.002, and an initial learning rate of 0.04, using one run of cosine annealing with a minimum learning rate of 0.04/32.

Using this robust teacher, we trained a student network using standard distillation, $f_{SD}$. We used SGD with a momentum of 0.9, a temperature of 4, a weight decay of 0.001 and made use of one run of cosine annealing with an initial learning rate of 0.04, falling to a minimum of 0.04/16. We use a loss mixing weight of $\alpha = 0.5$.

Next, we used the same robust teacher and trained a student network now with ARD, $f_{ARD}$. For the maximisation step during training, we again use the PGD, using 10 steps with a step size of 0.05 and an $\epsilon$ of 0.2. Again, we use SGD with a momentum of 0.9, a temperature of 2, and a weight decay parameter of 0.001. Finally, we again use one run of cosine annealing with an initial learning rate of 0f 0.04, decaying to a minimum learning rate of 0.04/32. We use a loss mixing weight of $\alpha = 0.5$.

Using the same robust teacher $f_{RT}$, we train a student network using RSLAD $f_{RSLAD}$. For the maximisation step during training, we again use the PGD, using 10 steps with a step size of 0.05 and an $\epsilon$ of 0.2. Again, we use SGD with a momentum of 0.9, a temperature of 2, and a weight decay parameter of 0.001. We again use one run of cosine annealing with an initial learning rate of 0f 0.04, decaying to a minimum learning rate of 0.04/32. We use a loss mixing weight of $\alpha = 0.5$.

Under FD, we trained a student network, $f_{FD}$, using the robust teacher described above. For the maximisation step during training, we again use the PGD, using 10 steps with a step size of 0.05 and an $\epsilon$ of 0.2. Following the maximisation step, we use SGD with a momentum of 0.9, a temperature of 2, and a weight decay parameter of 0.001. We started with an initial learning rate of 0.04 and used one run of cosine annealing, with a minimum learning rate of 0.04/16. We use a loss mixing weight of $\alpha = 0.5$.

### A.2    Training Settings for Networks Used in Fashion MNIST Experiments

We train all models over 64 epochs using early stopping with a patience of 8 epochs based on the test accuracy. The teacher and student networks have {64, 32, 32, 32, 32, 16, 10} and {30, 30, 30, 30, 10} neurons per fully-connected layer.

The teacher, $f_{RT}$, was trained using adversarial training, utilising the PGD attacks with an $\epsilon$ of 8/255 and step size of 2/255 for 10 iterations. We used SGD with a momentum of 0.9, a weight decay factor of 0.002, and an initial learning rate of 0.04, using one run of cosine annealing with a minimum learning rate of 0.04/32.

Using this robust teacher, we trained a student network using standard distillation, $f_{SD}$. We used SGD with a momentum of 0.9, a temperature of 4, and a weight decay of 0 and used one run of cosine annealing with an initial learning rate of 0.01, falling to a minimum of 0.01/32. We use a loss mixing weight of $\alpha = 0.5$.

Next, we used the same robust teacher and trained a student network now with ARD, $f_{ARD}$. For the maximisation step during training, we again use the PGD, using 10 steps with a step size of 2/255 and an $\epsilon$ of 8/255. Again, we use SGD with a momentum of 0.9, a temperature of 2, and a weight decay parameter of 0.001. Finally, we again use one run of cosine annealing with an initial learning rate of 0f 0.01, decaying to a minimum learning rate of 0.01/8. We use a loss mixing weight of $\alpha = 0.5$.

Using the same robust teacher $f_{RT}$, we train a student network using RSLAD $f_{RSLAD}$. For the maximisation step during training, we again use the PGD, using 10 steps with a step size of 2/255 and an $\epsilon$ of 8/255. Again, we use SGD with a momentum of 0.9, a temperature of 2, and a weight decay parameter of 0. We again use one run of cosine annealing with an initial learning rate of of 0.02, decaying to a minimum learning rate of 0.02/32. We use a loss mixing weight of $\alpha = 0.5$.

Under FD, we trained a student network, $f_{FD}$, using the robust teacher described above. For the maximisation step during training, we again use the PGD, using 10 steps with a step size of 2/255 and an $\epsilon$ of 8/255. Following the maximisation step, we use SGD with a momentum of 0.9, a temperature of 2, and a weight decay parameter of 0.001. We started with an initial learning rate of 0.02 and used one run of cosine annealing, with a minimum learning rate of 0.02/32. We use a loss mixing weight of $\alpha = 0.5$.

## A.3 Training Settings for Networks Used in CIAFR-10 Experiments

We train all models over 128 epochs using early stopping with a patience of 12 epochs based on the test accuracy. Table 4 shows the network architectures of the teacher and student networks used within our CIFAR-10 experiments.

The teacher, $f_{RT}$, was trained using adversarial training, utilising the PGD attacks with an $\epsilon$ of 8/255 and step size of 2/255 for 10 iterations. We used SGD with a momentum of 0.9, a weight decay factor of 0.001, and an initial learning rate of 0.1, using one run of cosine annealing with a minimum learning rate of 0.01/32.

Table 4: CIFAR-10 teacher and student network architectures.

| Teacher Network | Student Networks |
|---|---|
| Conv(input=3, filters=8, kernel=3, stride=2) | Conv(input=3, filters=6, kernel=3, stride=2) |
| ReLU | ReLU |
| Pool(2, stride=2) | Pool(2, stride=2) |
| Conv(input=8, filters=16, kernel=3, stride=2, pad.=1) | Conv(input=6, filters=16, kernel=3, stride=2, pad.=1) |
| ReLU | ReLU |
| Conv(input=16, filters=32, kernel=3, stride=1) | Conv(input=16, filters=32, kernel=3, stride=1) |
| FC(input=128, output=128) | FC(input=128, output=64) |
| ReLU | ReLU |
| FC(input=128, output=64) | FC(input=64, output=32) |
| ReLU | ReLU |
| FC(input=64, output=10) | FC(input=32, output=10) |

Using this robust teacher, we trained a student network using standard distillation, $f_{SD}$. We used SGD with a momentum of 0.9, a temperature of 2, and a weight decay of 0.001 and used one run of cosine annealing with an initial learning rate of 0.1, falling to a minimum of 0.1/32. We use a loss mixing weight of $\alpha = 0.5$.

Next, we used the same robust teacher and trained a student network now with ARD, $f_{ARD}$. For the maximisation step during training, we again use the PGD, using 10 steps with a step size of 2/255 and an $\epsilon$ of 8/255. Again, we use SGD with a momentum of 0.9, a temperature of 2, and a weight decay parameter of 0.001. Finally, we again use one run of cosine annealing with an initial learning rate of of 0.1, decaying to a minimum learning rate of 0.1/32. We use a loss mixing weight of $\alpha = 0.5$.

Using the same robust teacher $f_{RT}$, we train a student network using RSLAD $f_{RSLAD}$. For the maximisation step during training, we again use the PGD, using 10 steps with a step size of $2/255$ and an $\epsilon$ of $8/255$. Again, we use SGD with a momentum of 0.9, a temperature of 2, and a weight decay parameter of 0. We again use one run of cosine annealing with an initial learning rate of of 0.1, decaying to a minimum learning rate of $0.1/16$. We use a loss mixing weight of $\alpha = 0.5$.

Under FD, we trained a student network, $f_{FD}$, using the robust teacher described above. For the maximisation step during training, we again use the PGD, using 10 steps with a step size of $2/255$ and an $\epsilon$ of $8/255$. Following the maximisation step, we use SGD with a momentum of 0.9, a temperature of 2, and a weight decay parameter of 0. We started with an initial learning rate of 0.1 and used one run of cosine annealing, with a minimum learning rate of $0.1/32$. We use a loss mixing weight of $\alpha = 1.0$ (appendix A.4 shows details of bounding results obtained through varying the value of $\alpha$ used during FD training).

### A.4   Ablation of Loss Weighting $\alpha$ for FD on CIFAR-10

Table A.4 shows an ablation study carried out to investigate the effect of the loss weighting $\alpha$ on the values EMPLB and FAITHUB. Here, we adopt the notation $f_{FD,\alpha}$ to denote a distilled FD student which was trained with loss weighting $\alpha$.

Table 5: *Faithfulness and relative calibration* - empirical lower bounds (EMPLB) and faithfulness upper bounds (FAITHUB) on MNIST and F-MNIST. *Lower is better.*

|  | $\epsilon$ | EMPLB | | | FAITHUB | | |
|---|---|---|---|---|---|---|---|
|  |  | $f_{FD,0.5}$ | $f_{FD,0.75}$ | $f_{FD,1.0}$ | $f_{FD,0.5}$ | $f_{FD,0.75}$ | $f_{FD,1.0}$ |
| CIFAR-10 | 4/255 | $0.199_{\pm 0.119}$ | $0.212_{\pm 0.120}$ | $\mathbf{0.187}_{\pm 0.096}$ | $0.525_{\pm 0.193}$ | $0.511_{\pm 0.196}$ | $\mathbf{0.474}_{\pm 0.182}$ |
|  | 8/255 | $0.265_{\pm 0.134}$ | $0.273_{\pm 0.137}$ | $\mathbf{0.245}_{\pm 0.114}$ | $0.907_{\pm 0.118}$ | $0.889_{\pm 0.132}$ | $\mathbf{0.864}_{\pm 0.153}$ |
|  | 12/255 | $0.333_{\pm 0.145}$ | $0.333_{\pm 0.150}$ | $\mathbf{0.302}_{\pm 0.129}$ | $0.993_{\pm 0.030}$ | $0.991_{\pm 0.035}$ | $\mathbf{0.989}_{\pm 0.038}$ |
|  | 16/255 | $0.400_{\pm 0.150}$ | $0.394_{\pm 0.159}$ | $\mathbf{0.359}_{\pm 0.142}$ | $0.999_{\pm 0.008}$ | $0.999_{\pm 0.008}$ | $\mathbf{0.999}_{\pm 0.009}$ |
|  | 20/255 | $0.462_{\pm 0.153}$ | $0.451_{\pm 0.166}$ | $\mathbf{0.416}_{\pm 0.151}$ | $1.000 \pm 0.000$ | $1.000_{\pm 0.001}$ | $\mathbf{1.000}_{\pm 0.001}$ |

