# OpenReview forum: "Faithful Knowledge Distillation"
_TMLR — Rejected by TMLR_

### Review · Reviewer_kZyx · 2023-06-19

**Summary Of Contributions:**

This paper investigates the confidence discrepancy between teacher and student models when facing adversarial samples in KD. The concept of a faithful imitator is introduced to discuss this problem and provide empirical lower and theoretical upper bounds. Based on this, faithful distillation loss is proposed to improve the consistency of student confidence with that of the teacher when dealing with adversarial samples. The effectiveness of the proposed bounds and loss is evaluated on the MNIST and Fashion-MNIST datasets, and compared with the previous distillation losses.


**Audience:**

Yes

**Claims And Evidence:**

Yes

**Requested Changes:**

Scaling the method to real-world datasets, e.g., CIFAR-10 or even harder datasets.

**Strengths And Weaknesses:**

# Strengths

- Providing a deeper understanding of the relationship between teacher and student models in knowledge distillation.
- Formalizing and analyzing the mismatch in confidence between teacher and student models when facing adversarial samples, with a clear formulation.

# Weaknesses

- The experiments are conducted only on small datasets and simple models. The paper acknowledges that the proposed method struggles to work with datasets like CIFAR-10, which is inconsistent with the motivation behind knowledge distillation typically being applied to larger datasets and models. Conducting experiments in such settings is not appropriate.
- Similar loss that introduces random perturbations to teacher and student inputs has already been discussed in [1]. A more thorough comparison and experimental analysis with [1] would better demonstrate the effectiveness of the Faithful Distillation loss.

---


[1] Beyer, L., Zhai, X., Royer, A., Markeeva, L., Anil, R., & Kolesnikov, A. (2022). Knowledge distillation: A good teacher is patient and consistent. In Proceedings of the IEEE/CVF Conference on Computer Vision and Pattern Recognition (pp. 10925-10934).

---

> ### Author Response · Authors · 2023-07-10
> **Authors' Reply to Reviewer kZyx**
>
> We would like to thank the reviewer for the time spent carefully reviewing our paper, and for the constructive comments provided.
>
> Firstly, we have added in the requested additional experiments on CIFAR-10 in a revised version of the paper that has been now been uploaded. This includes additional results in Tables 1 and 2 and additional experiment details at the start of Section 6 and in Appendices A3 and A4.
>
> In regard to the second bullet point in the weakness section, the authors in [1] use the same loss as introduced by Hinton et al. [2] except without the hard label term. This is effectively covered within the paper through SD (which is just the loss introduced by Hinton et al. [2]) or through RSLAD where no hard labels are used within the loss. Although the authors, as we do in this paper, consider the problem of knowledge distillation as function matching in some sense, the authors augment their inputs mainly through data augmentations such as cropping and do not consider adversarial examples or robustness specifically (they set out to investigate design choices that lead to effective standard distillation). Hence, the paper does not seem directly related to the goals and problems addressed within our paper.
>
> [2] Hinton, G., Vinyals, O., & Dean, J. (2015). Distilling the knowledge in a neural network. arXiv preprint arXiv:1503.02531.

---

### Review · Reviewer_A7Et · 2023-06-22

**Summary Of Contributions:**

This paper studies knowledge distillation considering adversarial robustness. More specifically, it focuses on the relative calibration of the student model w.r.t. to the teacher model. To address two questions about the distillation behavior between the student and teacher models, it proposes a faithful imitation framework (termed Faithful Distillation, FD) to discuss and evaluate the relative calibration of a student model. Experiments conducted on the MNIST and Fashion-MNIST datasets demonstrate the significance of using such a framework.

**Audience:**

Yes

**Broader Impact Concerns:**

none as of this time

**Claims And Evidence:**

Yes

**Requested Changes:**

Major:
1. Please refer to the previous Weaknesses part to respond and revise accordingly.
2. As previously mentioned, please discuss the missing related work [2] that also discusses the disagreement between teacher and student in adversarial knowledge distillation.
3. The loss function (i.e., Eq.(12)) is quite similar with TRADES[3], ARD[1],  RSLAD, and IAD[2], could the author also summarize them and discuss the loss objective design?

[3]. Zhang H, Yu Y, Jiao J, Xing E, El Ghaoui L, Jordan M. Theoretically principled trade-off between robustness and accuracy. In ICML 2019.

Minor:
1. For the adversarial disagreement examples, it could be better to use more visualization or illustration to show the overview of each case considering the classification results and agreement/disagreement of teacher and student models under adversarial distillation.



**Strengths And Weaknesses:**

Strengths:
1. This research work studies the adversarial robustness of the distilled student model, which has research significance for obtaining adversarially robust models.
2. The studied relative calibration of a student model w.r.t. its teacher model in the term of adversarial robustness provides a more comprehensive understanding of the distillation behavior under the conventional adversarial distillation framework [1].
3. The experiments on MNIST and F-MNIST empirically demonstrate the effectiveness of the proposed FD in term of relative calibration.

Weaknesses:
1. Before introducing the first important research question (i.e., can we find dataset examples where the teacher and ...), it is unclear what reason naturally motivates us to study the question. In other words, why do we need to know the disagreement under the small perturbations as the research work claimed? (also for the second research question). The current motivation illustrated by the example of Figure 1 is not so strong and intuitive for the significance of researching the questions.
2. Another concern about the motivation is, why we expected that the student model be small differences in confidence with its teacher. It needs more discussion and explanation before the statement at the end of page 1. Considering a missing case, previous related work [2] also discusses the disagreement of teacher and student models in adversarial distillation, where the adversarial example that the teacher can not correctly classify, would the student model also be expected to be confident as the teacher model? It would be better to discuss more for completed consideration under the research questions.
3. Figure 1 could be more intuitive in showing the research problem about standard distillation, and the current presentation lacks the important motivation or statement for the significance of faithful distillation. For the latter problem, would faithful distillation also be important beyond adversarial distillation? It would be an interesting question to discuss.
4. For the experimental part, it is wondering whether the method is also effective and whether the properties of FD also hold on the benchmarked datasets (i.e., CIFAR-10, CIFAR-100) or more large-scale datasets for adversarial robustness.

[1]. Goldblum, M., Fowl, L., Feizi, S., & Goldstein, T. (2020, April). Adversarially robust distillation. In AAAI 2020.

[2]. Zhu, J., Yao, J., Han, B., Zhang, J., Liu, T., Niu, G., ... & Yang, H. Reliable Adversarial Distillation with Unreliable Teachers. In ICLR 2022.

---

> ### Author Response · Authors · 2023-07-10
> **Authors' Reply to Reviewer A7Et**
>
> We would like to thank the reviewer for the time spent carefully reviewing our paper, and for the constructive comments provided.
>
> Firstly, we have added in the requested additional experiments on CIFAR-10 in a revised version of the paper that has been now been uploaded. This includes additional results in Tables 1 and 2 and additional experiment details at the start of Section 6 and in Appendices A3 and A4.
>
> Regarding the investigated research problem, the motivation to study such a problem stems from the fact that we wish to quantify how much our student is able to match the behaviour of its teacher. Specifically, if the student receives some perturbation to a particular input, we wish to see how closely the student is able to behave similarly to its teacher. With a framework that could measure this degree of imitation, one could more confidently deploy such a student in safety critical environments as if we are confident in the robustness, performance and safety of its teacher, then for a student that is a faithful imitator, that behaves similarly to its teacher, we can also be more confident in its safe deployment in resource constrained, safety crucial environments that necessitate model compression. We will add this to the introduction in the final version of the paper before the problem statement.
>
> We thank the reviewer for bringing [2] to our attention. We will add this to our related work section in the final version of the paper. In [2], the authors note the increasingly unreliable nature of fully trusting the teacher’s clean outputs during adversarial training (i.e, when trying to match a student on a perturbed image to the teacher’s outputs on the clean image). However, in [4],  the authors note from their experiments that specifically looking at the attention maps of RSLAD and IAD trained students, their RSLAD students are empirically observed to better imitate its teacher (i.e. to better match its confidences). Therefore investigating RSLAD is more directly relevant for the main purpose of our paper, which is to empirically investigate and verifiably introduce methods of measuring the degree of agreement between a student and its teacher. We will add in the discussion on the relation between RSLAD and IAD our background section.
>
> In regard to why we expect small differences in confidence between a teacher and its student under perturbation: if a student is trained solely to match its teacher on clean inputs for example, using say the loss introduced by Hinton et al. [5], there are no guarantees that the student would match its teacher in terms of confidence for perturbed examples (the student is simply encouraged to match its teacher at clean data points). Indeed, our empirical findings using our framework (see Table 2, specifically $f_{SD}$), show that this is demonstrably the case both empirically and verifiably so.
>
> Concerning making figure 1 more intuitive, we will add the additional information within the figure highlighting the smaller difference in confidence between the teacher and student in the diagram and indicating the reduced number of adversarial disagreement examples over standard distillation in the final version of the paper. This should make the connection between the figure and the motivation discussed above more clear.
>
> Looking at the uses of FD beyond adversarial distillation, FD could indeed be important but such cases would require further investigation which is beyond the scope of this work which focuses particularly on adversarial distillation.
>
> As highlighted by the reviewer, we compare the different loss function designs. Both ARD and RSLAD aim to train students on perturbed inputs to match their teachers on clean inputs. IAD, in a similar way to TRADES, utilises an additional introspection loss term that aims to encourage the student on a perturbed input to match itself on the clean input increasingly during training. All three methods generate this adversary by maximising a cross-entropy loss between the student under perturbation and the correct label within an $\epsilon$ ball surrounding a given image. FD on the other hand, tries to match the student and teacher whilst both are perturbed, with the perturbation found by maximising the difference in distributions of the teacher and student under perturbation as measured by the KL divergence. This has the aim of adding an inductive bias to the training process where we wish to actively encourage the student to match its teacher within an $\epsilon$ ball of perturbations. Will will add this discussion during our introduction to the FD loss.
>
> [4] Zi, B., Zhao, S., Ma, X., & Jiang, Y. G. (2021). Revisiting adversarial robustness distillation: Robust soft labels make student better. In Proceedings of the IEEE/CVF International Conference on Computer Vision (pp. 16443-16452).
>
> [4] Hinton, G., Vinyals, O., & Dean, J. (2015). Distilling the knowledge in a neural network. arXiv preprint arXiv:1503.02531.

---

### Review · Reviewer_C2ns · 2023-06-25

**Summary Of Contributions:**

This article introduces a novel approach called "faithful knowledge distillation" that tackles the issue of relative calibration between the student network and teacher network, specifically regarding soft confidences. The authors explore two main research questions: (1) whether disagreements between the teacher and student models occur near correctly classified samples, and (2) whether the student network can achieve similar confidence levels as the teacher network for dataset samples. By providing an illustrative example, the authors demonstrate that the conventional knowledge distillation techniques fail to produce smooth differences in confidence between the teacher and student. As a result, the authors propose the concept of a "faithful imitator" that aims to minimize feature distance within a ball for dataset samples. The proposed concepts are supported by theoretical guarantees. On the algorithmic side, the formulation is straightforward and akin to the normal ARD formulation.



**Audience:**

Yes

**Claims And Evidence:**

Yes

**Requested Changes:**

Larger scale experiments; explanation on why the delta should be the same for teacher and student.

**Strengths And Weaknesses:**

Strength:

- The setting is well motivated and the solution seems to be easy
- The paper provides theoretical understanding for the proposed algorithm
- The experimental results provide substantial evidence, demonstrating the superiority of the proposed method. In particular, the method exhibits comparable performance in terms of individual model robust accuracy while achieving higher empirical distillation agreement.

Weakness:

- The experiment is only conducted on MNIST and FashionMNIST, which are relatively small datasets. Conducting experiments on larger datasets such as CIFAR-10 would provide a more comprehensive evaluation and demonstrate the superiority of the proposed methods on a larger scale. This is also mentioned by the authors in the limitation section.



- The performance gap between the proposed algorithms and the baseline is not significant even on MNIST and FashionMNIST. In Table 1, it would be helpful to provide the standard deviation for all the numbers to assess the consistency of the results. Additionally, in Table 2, it is evident that the proposed algorithms do not consistently outperform the baseline. Furthermore, the large standard deviations observed indicate high variability in performance. In certain cases, the coefficient of variation exceeds 1, suggesting substantial fluctuation in the results.



- Regarding the method, one question arises: Why do we assume that the delta vector is the same for both the teacher and the student network? Would the results potentially improve if we decouple them and allow for different delta vectors?

---

> ### Author Response · Authors · 2023-07-10
>
> We would like to thank the reviewer for the time spent carefully reviewing our paper, and for the constructive comments provided.
>
> Firstly, we have added in the requested additional experiments on CIFAR-10 in a revised version of the paper that has been now been uploaded. This includes additional results in Tables 1 and 2 and additional experiment details at the start of Section 6 and in Appendices A3 and A4.
>
> Regarding the use of standard deviation in Table 1, in this table we report measures of robustness, specifically the robust accuracy of the individual models and the empirical distillation agreement which can be thought of as a measure of the robustness of the distillation process for a particular student. In particular, these are simply calculated as the percentage of adversarial examples or adversarial disagreement examples within a dataset. As a result, both are statistics of the full dataset which is why we haven’t reported the standard deviation here but have, say, in Table 2.
>
> Regarding the performance of FD in comparison to the baseline, although it is true that the proposed FD student does not output perform RSLAD in Table 2 on F-MNSIT if we look solely at EmpLBs, we see that the FD student has tighter FaithUBs. This is more desirable due to being a true, verified upper bounds whereas, as Table 3 highlights, EmpLB is empirical and therefore not reliable as a measure when used alone (due to say being dependent on the attack method used or say due to the number of attack iterations used).
>
> The large deviations noted by the reviewer are an artefact of the bounding method. However, this in itself is not necessarily a bad thing. For larger values of $\epsilon$ we see the mean of the FaithUB values for all students can be quite large. But, it is worth noting that EmpLB and FaithUB are effectively capped at 1. Hence, a larger standard deviation is desirable and indicates more images with lower differences in confidences. This is further evidenced in Figure 2, where we notice accumulations of images in the lower sections of the histograms away from the mean for a particular value of $\epsilon$.
>
> The motivation for using the same delta vector for both the teacher and student is due to the goal of training a student that best imitates its teacher’s behaviour. The goal is to encourage our student to match its teachers' soft confidences within an $\epsilon$ ball surrounding a given image at the same perturbed point. Hence, the goal is to find a perturbation that causes the maximum difference in confidences within this region and minimise this difference between the student and teacher when training using the FD loss. Decoupling or using a different delta vector would diverge from the goal of training a student to directly imitate the teacher. In particular, comparing $f_T(x+\epsilon_1)$ with $f_S(x+\epsilon_2)$ is not meaningful within this context since the input to the two networks will be different.

---

### Decision · Action_Editors · 2023-07-30

**Recommendation:** Reject

**Comment:**

This paper studies knowledge distillation considering adversarial robustness. The authors focus on the relative calibration of the student model wrt the teacher model, and propose a faithful imitation framework called faithful distillation.

It received 1 Leaning Accept and 2 Leaning Reject recommendations. The research setting of this paper is well-motivated, with comprehensive explanations. The method has a theoretical understanding to show its rationality. However, all the reviewers have shared one major concern that the experiments are conducted only on very small datasets and simple models. During author response, the authors have added results on CIFAR-10. However, this is not enough, and the editor agrees on this. The considered models are only very simple CNN architectures, it is highly suggested to test the proposed method on modern image backbones, such as ResNet, etc. Furthermore, CIFAR-10 is also a pretty small dataset, it is highly suggested to test the proposed method on CIFAR-100 and ImageNet as well.

Overall, the experiment scope is the major concern. Only conducting experiments on MNIST, Fashion MNIST, and CIFAR-10 with simple image backbones is hard for the editor to support acceptance of the paper; therefore, the editor decided to recommend rejection by the end.

**Audience:**

Researchers in the field of adversarial robustness and knowledge distillation would be interested.

**Claims And Evidence:**

The claims in the submission are supported by comprehensive explanations and some theoretical understanding. However, the experiments are not convincing enough.

**Resubmission Of Major Revision:**

The authors may consider submitting a major revision at a later time.